# Evolving parsec-scale radio structure in the most distant blazar known

Tao An [1]*, Prashanth Mohan [1], Yingkang Zhang [1,2], Sándor Frey [3], Jun Yang [4], Krisztina É. Gabányi [5,3,6], Leonid I. Gurvits [7,8], Zsolt Paragi [7], Krisztina Perger[6,3] & Zhenya Zheng[1]

Blazars are a sub-class of quasars with Doppler boosted jets oriented close to the line of sight, and thus efficient probes of supermassive black hole growth and their environment, especially at high redshifts. Here we report on Very Long Baseline Interferometry observations of a blazar J0906 + 6930 at $z = 5.47$, which enabled the detection of polarised emission and measurement of jet proper motion at parsec scales. The observations suggest a less powerful jet compared with the general blazar population, including lower proper motion and bulk Lorentz factor. This coupled with a previously inferred high accretion rate indicate a transition from an accretion radiative power to a jet mechanical power based transfer of energy and momentum to the surrounding gas. While alternative scenarios could not be fully ruled out, our results indicate a possibly nascent jet embedded in and interacting with a dense medium resulting in a jet bending.

[1] Shanghai Astronomical Observatory, Key Laboratory of Radio Astronomy, 80 Nandan Road, 200030 Shanghai, China. [2] University of Chinese Academy of Sciences, 19A Yuquan Road, Shijingshan District, 100049 Beijing, China. [3] Konkoly Observatory, CSFK, Konkoly Thege Miklós út 15-17, H-1121 Budapest, Hungary. [4] Department of Space, Earth and Environment, Chalmers University of Technology, Onsala Space Observatory, SE-439 92 Onsala, Sweden. [5] MTA-ELTE Extragalactic Astrophysics Research Group, Pázmány Péter sétány 1/A, H-1117 Budapest, Hungary. [6] Department of Astronomy, Eötvös Loránd University, Pázmány Péter sétány 1/A, H-1117 Budapest, Hungary. [7] Joint Institute for VLBI ERIC (JIVE), Postbus 2, NL-7990 AA Dwingeloo, The Netherlands. [8] Department of Astrodynamics and Space Missions, Delft University of Technology, Kluyverweg 1, 2629 HS Delft, The Netherlands. *email: antao@shao.ac.cn

**M**echanisms for the formation and rapid growth of supermassive black holes (SMBHs) in the early Universe remain debatable[1] and have a complex connection with the evolution of their host galaxies through feedback[2]. The discovery of quasars at redshift $\gtrsim 6$ (refs. [3,4]) indicates that SMBHs as heavy as ~$10^9$ solar masses ($M_\odot$) have already existed when the Universe was at about a tenth of its current age. Spectroscopic surveys have largely enabled the discovery of high-redshift galaxies and quasars[3], paving the way for deeper optical, infrared and radio follow-up observations[5–8]. Currently there are more than 200 quasars discovered above redshift of 5.7 (ref. [7]). High-redshift blazars are useful probes of the early Universe owing to their Doppler-beamed emission which makes them among the brightest sources. As their jet power scales with the mass of the SMBH[9], blazars shed light on the cosmic evolution of massive black holes[10,11]. Jetted but misaligned quasars (i.e. larger inclination angle towards the observer's LOS (line of sight); not strongly beamed) are expected to outnumber blazars at a given redshift by a factor ~$\Gamma^2$ (where $\Gamma$ is the jet bulk Lorentz factor) rendering the occurrence of blazars rare at high redshifts[9]. An increasing sample size can help probe their expected number density and luminosity evolution over cosmic time[12], essential inputs for planning future surveys. This information can help to study the formation of SMBHs in the early Universe[11,13], active galactic nucleus (AGN) activity, the interaction of jets with the surrounding interstellar medium (ISM) and AGN feedback influencing the evolution of the host galaxy[2].

The source J0906 + 6930 ($z = 5.47$), identified as a blazar[14] remains the farthest yet in its class of objects. Unravelling the jet structure of high-redshift blazars requires extremely high resolution. It has a prominent pc-scale core-jet structure, unravelled by Very Long Baseline Array (VLBA) observations at 15 GHz[14,15]. The archival 15 GHz and new 22 GHz data obtained with the Korean VLBI network and the Japanese VLBI Exploration of Radio Astrometry (KaVA) arrays confirm a core-jet structure with a projected size of ~5 pc, extending to the southwest direction[16].

Here, we report the measurement of proper motion and linear polarisation in the parsec-scale jet of this high-redshift blazar. We use new 15-GHz data observed with the VLBA in 2017 and 2018, archival VLBA data obtained in 2004–2005 (see details in Supplementary Table 1) and the flux densities reported by the 40 m telescope at the Owens Valley Radio Observatory (OVRO) to explore the evolution of the source morphology and infer its physical characteristics. The jet parameters (lower proper motion and bulk Lorentz factor) are inclined to support a less powerful jet, compared with the general blazar population. The jet interacts with the surrounding interstellar medium resulting in a jet bending and polarised emission.

## Results

The new 15-GHz VLBA images are shown in Fig. 1d, e and the archival images in Fig. 1a–c. The noise in the image from the 2017 observation is a factor of 7–16 lower than those obtained during 2004–2005. All image parameters (beam size, peak brightness and rms noise) are presented in Supplementary Table 2. The peak brightness of the 2017 and 2018 images is ~3 times lower than 13 years ago, consistent with the declining flux density as indicated from the long-term 15 GHz light curve based on single-dish monitoring at the OVRO (see Supplementary Fig. 1).

A compact core-jet structure is present in all images. The shape of the elliptical Gaussian model indicates that the core region (C) is a blend of the optically thick (at 15 GHz) jet base and an inner section of the optically thin jet. The major to minor axis ratio of C ranges between 3.7 and 13.5 with a northeast–southwest elongation. The jet component J1 is ~0.9 mas away from the core at a position angle of ~$-138°$. In the highest-resolution 2005 image[16] and the new 2017 image, a sharp (>90°) jet bending is seen from the southwest to the south at the location of J1. The fainter component J2 is at the end of the pc-scale jet, about 1 mas south of the core. The same bent jet morphology is seen at lower frequencies[17], up to ~2 mas from C.

Apparently abrupt changes in jet direction on pc scales in blazars are frequently observed. In most cases this implies a slight change of direction in the jet which points very close to our LOS. The jet bending itself may indicate a low-pitch angle helical motion[18], like, e.g., in the well-studied blazar 1156 + 295 (ref. [19]). Alternatively, the jet bending may also result from interaction with massive clouds in the ISM, e.g. dense clouds in the broad or narrow line regions[20,21]. In the present case of J0906 + 6930, we find no indication for helical motion, for example, there is no noticeable variation in the core position angle or in the shape of the optically thick jet base (i.e. the fitted core component in Table 1), and there are no significant periodic variations seen in the light curve (Supplementary Fig. 1). There is however support for possible jet–ISM interaction, evidenced by the relatively high levels of linear polarisation observed near the jet bending (see discussion below). The density contrast between the material in the jet and that external to it is >9 (see Methods: jet and ISM properties), thus suggesting a relatively lighter jet susceptible to interaction and bending owing to a relatively denser medium. Assuming a momentum balance across the jet–ISM interaction interface, the ISM number density is estimated to be $n_e \gtrsim 26.6$ cm$^{-3}$.

The position of J1 (projected distance ~5.3 pc away from the core) is nearly stationary between 2004 and 2018, consistent with a jet beam encountering dense surrounding ISM. This is also supported by the increasing flux density at J1, which represents a standing shock where the material and magnetic fields near the jet

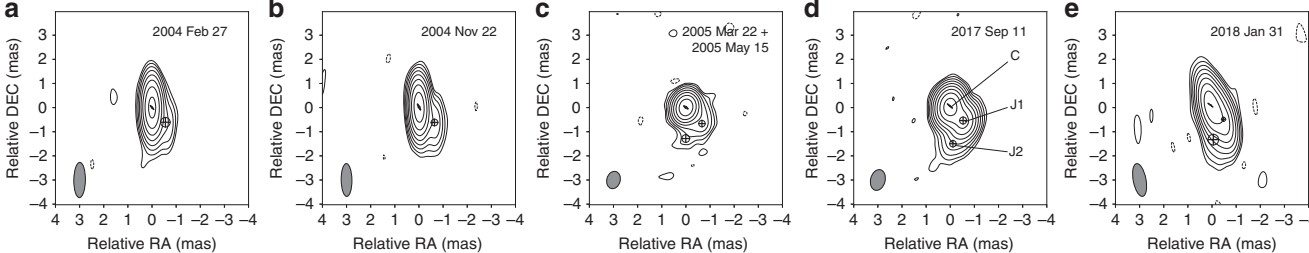

**Fig. 1 Radio morphology of J0906 + 6930 derived from VLBA observations at six epochs.** The data of 22 March 2005 and 15 May 2005 are combined to create a single image (**c**). Symbols (elliptical and circular Gaussian) represent the model fitting of emission components. Detailed imaging parameters are listed in Supplementary Table 2. The noise rms is 0.18, 0.19 and 0.16 mJy beam$^{-1}$ for the 2004–2005 epoch images (**a**, **b** and **c**, respectively) and is much smaller at 0.034 and 0.057 mJy beam$^{-1}$ for the 2017 and 2018 epoch images (panels **d** and **e**, respectively). The core C is the brightest and most compact. Two jet components, marked as J1 and J2, are detected within 1.5 mas away from the core. The contours increase by a factor of 2. The grey ellipse at the bottom left corner of each panel represents the full-width at half-maximum (FWHM) of the restoring beam.

**Table 1 Model fitting parameters.**

| Epoch (yyyy mm dd) | Comp | $S_{total}$ (mJy) | $D_{maj}$ (mas) | $D_{min}$ (mas) | $\varphi$ (°) | $R$ (mas) | PA (°) | $T_B$ (×10¹⁰ K) |
|---|---|---|---|---|---|---|---|---|
| (1) | (2) | (3) | (4) | (5) | (6) | (7) | (8) | (9) |
| 2004 02 27 | C | 119.5 ± 6.3 | 0.206 ± 0.005 | 0.041 ± 0.001 | 40.9 ± 0.9 | – | – | 47.5 ± 3.0 |
| | J1 | 7.2 ± 0.7 | 0.396 ± 0.051 | – | – | 0.825 ± 0.068 | 222.2 ± 1.1 | – |
| 2004 11 22 | C | 127.3 ± 6.6 | 0.269 ± 0.002 | 0.065 ± 0.002 | 28.1 ± 0.1 | – | – | 24.4 ± 1.5 |
| | J1 | 8.7 ± 0.8 | 0.279 ± 0.027 | – | – | 0.906 ± 0.064 | 225.0 ± 0.3 | – |
| 2005 03 22 & | C | 122.7 ± 6.4 | 0.209 ± 0.001 | 0.067 ± 0.002 | 55.3 ± 0.1 | – | – | 31.0 ± 1.8 |
| 2005 05 15 | J1 | 8.4 ± 0.6 | 0.267 ± 0.007 | – | – | 0.944 ± 0.052 | 224.7 ± 0.2 | – |
| | J2 | 1.9 ± 0.3 | 0.321 ± 0.053 | – | – | 1.290 ± 0.061 | 179.8 ± 1.3 | – |
| 2017 09 11 | C | 43.4 ± 2.3 | 0.260 ± 0.001 | 0.032 ± 0.001 | 49.4 ± 0.1 | – | – | 18.0 ± 1.1 |
| | J1 | 20.4 ± 1.1 | 0.291 ± 0.001 | – | – | 0.814 ± 0.049 | 222.2 ± 0.1 | – |
| | J2 | 1.0 ± 0.1 | 0.270 ± 0.033 | – | – | 1.568 ± 0.053 | 184.6 ± 0.1 | – |
| 2018 01 31 | C | 41.8 ± 2.2 | 0.249 ± 0.004 | <0.034 | 52.2 ± 1.3 | – | – | >17.1 |
| | J1 | 18.0 ± 1.0 | 0.170 ± 0.001 | – | – | 0.801 ± 0.071 | 222.8 ± 0.1 | – |
| | J2 | 1.4 ± 0.2 | 0.444 ± 0.079 | – | – | 1.435 ± 0.076 | 184.8 ± 0.9 | – |

Parameters are derived from modelled Stokes LL images. Column (1) presents the observation epoch. Column (2) gives the label of the VLBI components. Column (3) presents the integrated flux density of all VLBI components. Columns (4) to (5) give the major and (in case of ellipticals) the minor axis sizes (FWHM) of the fitted Gaussian models. Column (6) is the position angle of the major axis of Gaussian, measured from north to east. The data from 22 March 2005 and 15 May were combined before model fitting. Columns (7) and (8) give the radial distance $R$ of components with respect to the core, and the position angle measured from north to east. Column (9) lists the calculated brightness temperature of the core. For the unresolved core, a maximum size is estimated, thus the lower limit of $T_B$ is given

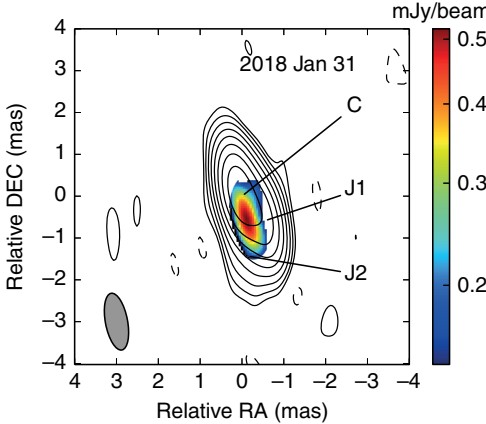

**Fig. 2 Linear polarisation image (coloured scale) of J0906 + 6930.** The images are derived from the 15-GHz VLBA observation on 31 January 2018. The core is denoted by C and jet components by J1 and J2. The contours represent Stokes I intensity, same as Fig. 1e. The coloured scale denotes the strength of the linear polarisation. The grey-shaded ellipse in the bottom left corner is the restoring beam. The peak of the polarised intensity, ~0.6 mJy beam⁻¹, is about 0.8 mas southwest of the total intensity core. The maximum fractional polarisation is ~10% appearing at the southernmost of the polarised component. The core region is weakly polarised. This is the only polarisation measurement in a radio-loud quasar at redshift >5 so far.

head are substantially compressed, resulting in an increased synchrotron emission[20]. This is indeed manifested in the polarisation image obtained from the 2018 epoch shown in Fig. 2. The polarised emission peaks at 0.8 mas south of the core and is aligned in the direction between J1 and J2. The maximum fractional polarisation is ~10%, and the peak polarised intensity is ~0.6 mJy beam⁻¹ (about 10 times above the rms noise). This is the initial polarisation measurement in a jet of a radio-loud quasar at redshift > 5. In contrast, the core is weakly polarised (<3σ). This is similar to the blazar CTA 102 (z = 1.037) observed at 43 GHz (rest frame frequency of ~88 GHz) which indicates a relatively low linear polarisation in the core with a remarkable increase of the polarisation intensity in the pc-scale jet, attributed to either a jet–ISM interaction or to preferential locations along a helical jet[22]. Statistical studies of radio polarisation in blazars

indicate cores with relatively low polarisation fractions of 2–3 % among sub-samples consisting of low and high synchrotron peaked sources[23]. If the synchrotron peak is associated with emission from the region downstream of a transverse shock, the polarisation near the core should be high owing to the shock passage resulting in an ordered magnetic field[24]. However, the low or near absent polarisation in the core of J0906 + 6930 indicates that the magnetic field could be turbulent or tangled.

A radio spectrum with a peak between 40 and 60 GHz (rest frame frequency)[16] and a weakly polarised core are characteristic of high-frequency peakers[25], where a higher peak frequency due to synchrotron self absorption is a consequence of a younger jet based on self-similar models of hot-spot expansion[26], consistent with the above inference from polarisation. A revised black hole mass of $4.4 \times 10^7 M_\odot$ is obtained if we use scaling relations appropriate to a Doppler-beamed jetted source. The mass is consistent with the expectation from a population of radio galaxies characterised with GHz peaked radio spectrum; together with the young AGN scenario, these are indicative of an ongoing transition in AGN feedback and an evolving black hole (see Methods black hole mass). Based on the location of the synchrotron peak frequency $\nu_s$ in the spectral energy distribution, blazars are classified into low synchrotron peaked if $\nu_s < 10^{14}$ Hz (in the infrared), intermediate synchrotron peaked if $10^{14}$ Hz $< \nu_s < 10^{15}$ Hz (optical–ultra-violet) and high synchrotron peaked if $\nu_s > 10^{15}$ Hz (in the X-rays). J0906 + 6930 is identified as a low synchrotron peaked blazar with $\nu_s \sim 10^{12}$ Hz[27]. Such sources are characterised by energetic jets with relatively highly superluminal (apparent) jet components. However, as this is not the case for J0906 + 6930 based on the low bulk Lorentz factor, slower component proper motion and a possible tangled or turbulent magnetic field, the jet is likely developing and nascent. Similarly, the recently discovered radio galaxy TGSS J1530 + 1049 at $z = 5.72$ also has a compact structure resembling a radio galaxy in an early evolutionary phase[28]. An alternative scenario of the J0906 + 6930 radio structure involving a re-started jet tracing a path swept by the past jet activity can not be fully ruled out from the present observation. The implications of AGN jet activity on SMBH growth in the early Universe and additional alternative scenarios enabling the jet structure are discussed in Methods: black hole mass. To check the latter picture, further high-sensitivity radio interferometric observations are necessary to search for relic emission structure on 100 mas scales.

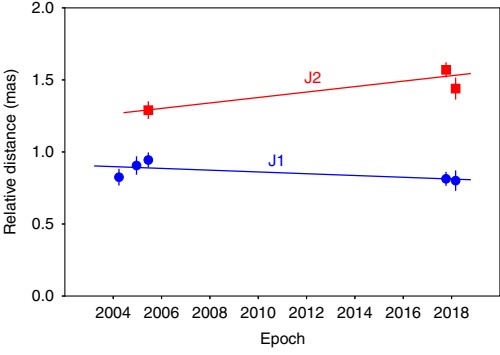

**Fig. 3 Radial distance of J1 and J2 as a function of observing time.** The straight lines (blue line for J1 and red line for J2) represent a linear regression fit to infer the jet proper motion. The denoted error bars on each point are the $1\sigma$ errors (see Table 1). That gives $\mu$(J1) = $-0.006 \pm 0.004$ mas yr$^{-1}$, and $\mu$(J2)=$0.019 \pm 0.006$ mas yr$^{-1}$.

At $z > 4.5$, about 50 quasars have been detected in radio bands with 30 of them being VLBI imaged[5]. As the sources are mostly dominated by compact single components with flux densities <20 mJy (at GHz observing frequencies), proper motion measurements face challenges owing to the relative scarcity, faintness and a requirement for long time gaps between epochs for a definitive estimate due to cosmological time dilation. Successful measurements have been only made for two $z > 4$ blazars with the European VLBI Network (EVN) using only two epochs. These include J1026 + 2542 ($z = 5.27$) with a proper motion of 3.3–14.0 $c$ over 7 years[29] and J2134−0419 ($z = 4.33$) with a proper motion of 4.1 ± 2.7 $c$ over 16 years[30]. The compact and bright jet in J0906 + 6930 makes it suitable for the study of jet kinematics. Combining the archival and new data, the proper motion of J1 is −0.006 ± 0.004 mas yr$^{-1}$ (−0.8 ± 0.5 $c$, Fig. 3), consistent with a scenario involving jet–ISM interaction and the subsequent jet bending. The separation of component J2 shows a visible increase from 1.27 mas in 2005 to 1.58 mas in 2017/2018. The apparent proper motion of J2 is 0.019 ± 0.006 mas yr$^{-1}$ (2.5 ± 0.8 $c$, Fig. 3). These are the preliminary measurements for a $z > 4.5$ blazar based on data spanning more than two epochs and are consistent with (much lower than) a maximal proper motion of 0.09 mas yr$^{-1}$ expected in a highly beamed jet in an accelerated cosmological expansion (see Methods maximum proper motion). For a sample of 122 relatively lower-redshift ($z = 0.1$–3) radio-loud AGN, the median jet proper motion peaks at $\lesssim 5c$[31], with the low synchrotron peaked sources indicating the fastest speeds (upto ∼40 $c$) and high Doppler boosted teraelectronvolt (TeV) γ-ray emission. Although the estimate for J0906 + 6930 is consistent within the statistical expectation, the apparent jet speed is significantly lower than the expected value for a low synchrotron peaked blazar.

Doppler beaming in the relativistic jet can cause the core brightness temperature ($T_B$) to exceed the theoretical limits set by equipartition between the kinetic and magnetic energy densities, assumed to be $T_{B,eq} = 5 \times 10^{10}$ K[32]. The $T_B$ of the J0906 + 6930 core is $(30.2 \pm 4.0) \times 10^{10}$ K with a consequent Doppler factor $\delta = T_B/T_{B,eq}$ of 6.1 ± 0.8 (see Methods Doppler boosting parameters). From the inferred apparent velocity and Doppler factor, the bulk Lorentz factor $\Gamma = 3.6 \pm 0.5$ and inclination angle towards the observer LOS $\theta = 6.8° \pm 2.2°$. These values are consistent with estimates from the parametric modelling of the spectral energy distribution[27]. The lower Lorentz factor and slower jet component are consistent with the less powerful nature of the J0906 + 6930 jet (jet/Eddington luminosity ∼0.004, see Methods: jet and ISM properties). The relatively less powerful jet in addition to clues from the radio spectrum and polarisation point to its possible nascent

nature. A prominent disk emission is inferred from the modelling of the spectral energy distribution of this and three other high-$z$ blazars[27]. This coupled with a relatively less powerful jet luminosity marks a possible transition in this source between the accretion or quasar mode and the onset of the jet or radio mode. In the former, the accretion energy can radiatively drive (momentum transfer) surrounding gas to galactic scales, and in the latter, a powerful jet can transfer mechanical energy acting to heat up the gas at galactic and cluster scales[2].

The present VLBI data thus characterise J0906 + 6930 as a high-redshift blazar with a nascent jet embedded in a dense medium causing the pc-scale jet–ISM interaction and the consequent jet bending. These represent the initial results of ongoing investigations on high-resolution imaging of a sample of high-redshift blazars. Further simultaneous multi-frequency VLBI observations can help constrain the magnetic field strength and electron density from Faraday rotation measurements. The next generation VLBI facilities (e.g. Square Kilometre Array VLBI programme[33]) are more sensitive to detect much weaker high-redshift blazars, thus advancing our understanding of the co-evolution of SMBHs and host galaxies in the early Universe.

## Methods

**VLBI observations.** The compact prominent jet and an elapsed time longer than 10 years since the earlier observations make J0906 + 6930 a promising source for continued monitoring of the evolution of the jet structure. We additionally compiled and reduced archival VLBA data in this analysis. The observational setup for each observation is presented in Supplementary Table 1.

We conducted VLBA 15 GHz observations on 11 September 2017 and 31 January 2018. All ten VLBA telescopes were requested in the proposals BZ068 and BZ071. However, due to unfavourable weather conditions and maintenance time, the Saint Croix (SC) telescope did not participate in the observations, resulting in relatively lower resolution in east–west direction in the 2017 and 2018 images compared to those obtained from the full VLBA in 2004. To enable a complete analysis of the pc-scale jet evolution in J0906 + 6930, we obtained all available 15-GHz VLBA data from the NRAO Archive (https://archive.nrao.edu/).

For the BZ068 observation, the data were recorded in four baseband channels, each with 64 MHz bandwidth. Each of the left-handed circular polarisation (LCP) and right-handed circular polarisation (RCP) occupies two channels. A 2-bit sampling resulted in a total data rate of 1 gigabit per second (Gbps). Phase referencing was not required as the source J0906 + 6930 itself was bright and compact enough to be used for fringe fitting. Except for a few scans which were used on the fringe finder (NRAO 150 and 3C 84) in the beginning and end of the observation run, the on-source time was about 400 min. The BZ068 observation thus led to a vast improvement in image quality (lowest image noise) compared to previous VLBA observations.

The main goal of the subsequent BZ071 observation was the detection of the linear polarisation in J0906 + 6930. The observation was carried out in full polarisation mode. The radio galaxy 3C 84 was used for instrumental polarisation calibration, and the quasar NRAO 150 for fringe searching and bandpass calibration. The recording settings are similar to those of BZ068 except that four 128-MHz baseband channels were used, resulting in a total data rate of 2 Gbps, twice that of BZ068. After observation, both datasets were correlated at the National Radio Astronomy Observatory in Socorro, USA, using the DiFX software correlator[34]. The post calibration and imaging were carried out at the China SKA Regional Centre prototype[35].

**Data reduction.** The correlated visibility data were imported into the NRAO Astronomical Image Processing System (AIPS) software package[36] for amplitude and phase calibration. We applied the standard calibration procedure of AIPS. The AIPS task APCAL was performed to calibrate the visibility amplitudes, using the antenna gains and system temperatures measured at each station during the observation. The atmospheric opacity was estimated based on the weather information recorded at each station and accounted for. The instrumental delay and global phase errors were then calibrated using the FRING task. This included a manual fringe fitting using NRAO 150 (∼10 Jy at this frequency in this epoch) as a calibrator to determine the delay offsets and phase errors between different sub-bands, and for application the solutions to all antennas. This was followed by running the global fringe fitting including all data to calculate and remove the global phase errors. Over 98% good solutions were achieved for both datasets. Then, the antenna-based bandpass functions were solved from the NRAO 150 data by using the task BPASS and applied to the visibility data. The bandpass shape across the broad 128–256 MHz baseband is corrected resulting in an increased dynamic range (the ratio of image peak to noise).

The calibrated data were averaged in each subband (each 64 MHz wide) and in time (2 s) and exported to external FITS files using the task SPLIT. The resulting single-source data file was imported into the Caltech Difmap package[37] to further calibrate residual phase errors. The hybrid mapping process consisted of several iterations of CLEAN and self-calibration. The final image was obtained after a few iterations of phase and amplitude self-calibrations, repeated by gradually reducing the solution intervals from 8 h to 1 min.

The raw correlated visibilities from the 2004 and 2005 observations were processed following the same procedure as discussed above using AIPS. The BG154B (22 March 2005) and BG154E (15 May 2005) data were combined owing to a close time separation after ensuring that there were no significant differences in the measured flux densities. This resulted in a high-resolution image with excellent (u,v) coverage in both N–S and E–W directions. This and the resulting improved sensitivity enabled the detection of the weak J2 jet component (see Fig. 1c), which was earlier not possible. All parameters including beam properties, peak brightness and noise rms for each image are presented in Supplementary Table 2.

**Model fitting and error estimation.** After self-calibration, the modelfit procedure (in Difmap) was used to fit the visibilities (at all epochs) with Gaussian brightness distribution models. An elliptical Gaussian model is used to fit the core while circular ones are used for the jet. In order to avoid positional and intensity offsets between LL and RR polarisations (a possible tiny difference), for the model fitting we used only LL cross-correlation products.

The fitted Gaussian models are shown as elliptical or circular shapes in Fig. 1. In the 2004 epochs, two components were detected: the core (C) and a southwest jet component (J1). Due to the improved sensitivity and better north–south (u,v) coverage, one more jet component J2 was detected at about 1.5 mas south of the core in the other three epochs. The core was unresolved on 31 January 2018, and its minor axis size was estimated as an upper limit by considering the restoring beam size and signal-to-noise level[38]. In all five epochs, although the (u,v) coverages and observation time ranges were different, the core component was elongated along the same northeast–southwest direction, with the position angles ranging between 28.1° and 55.3°, roughly within three times σ (rms). This is possibly a true feature or could arise as an artefact of the VLBA (u,v)-coverage pattern. It is worth noticing that the core component major axis always points towards J1 (see Table 1). These clues together indicate that the unresolved core region likely contains an opaque core (at this frequency) and an inner jet along the C–J1 direction. In the model fitting of the 11 September 2017 epoch (highest quality data), an emission component of ~0.6 mJy beam$^{-1}$ peaks between C and J1 in the residual image after removing these components, adding weight to the possible emergence of a newly generated, yet unresolved jet component.

The statistical errors from the fitted Gaussian models are rough estimates based on the signal-to-noise levels in the images[39]. In addition to the statistical error $\sigma_{S,\,sta}$, the true errors can comprise contributions from other factors resulting from the calibration error of flux density scale, incomplete (u,v) coverage of the interferometric array and the complex jet structure. An extra 5% of flux density calibration error, which is the typical value for the VLBA[40], is added to account for the calibration error $\sigma_{S,\,cal}$ of the visibility amplitude. Thus the uncertainty of the flux density is estimated as $\sigma_S = \sqrt{\sigma_{S,cal}^2 + \sigma_{S,sta}^2}$.

The uncertainty of the Gaussian component size is decomposed as the statistical error and the fitting error. The statistical error is $\frac{\theta_D}{SNR}$, where the SNR is the signal-to-noise ratio of the fitted Gaussian component, $\theta_D$ is the fitted Gaussian size $D_{maj}$ and $D_{min}$ listed in columns 4–5 of Table 1. The final size uncertainty is the quadratic mean of these two errors.

The positions of jet components are measured with respect to the core which is assumed to be stationary. The statistical error on the position is $\sigma_{p,sta} = \frac{\theta}{2*SNR}$, where $\theta$ is taken as the synthesised beam size (the full-width at half-maximum, FWHM). As mentioned above, the core component is likely mixed with inner jet emission. In reality, the peak emission of the core can be affected by intrinsic changes of the emission structure (for example, the ejection of a new jet component or a passing shock) and the goodness of the intensity distribution being fitted with a Gaussian function. All these factors may introduce an additional uncertainty to the reference point. We have fitted the core with several different models: a single elliptical Gaussian; one circular Gaussian component (the core) plus a point source (to represent the residual emission from the unresolved inner jet); two circular Gaussian components; two point sources. The discrepancy between the fitted core positions is used as the systematic error of the reference point $\sigma_{p,sys}$, which is added into the total error budget. The positional error of jet component can be expressed as $\sigma_p = \sqrt{\sigma_{p,sta}^2 + \sigma_{p,sys}^2}$. We found that the positional error of the brighter component J1 is dominated by the systematic error, while the statistical error contributes a significant fraction to the positional error of J2. The positional errors of J1 and J2 as well as the derived proper motions are tabulated in Supplementary Tables 3–5.

**Polarisation calibration.** The VLBA observing project BZ071 presented in this paper was designed primarily as the preliminary exploratory attempt to detect a pc-scale polarised emission from the source. As the observing period was only 2 h, the primary focus was on inferring the intensity and the location of the polarised

emission; calibration of absolute electric vector position angle (EVPA) was not attempted since it would decrease already short total exposition on the traget source. The unpolarised source 3C 84 was used as the instrumental polarisation (so-called D-term) calibrator.

The correlated visibilities were then imported into AIPS to calibrate the amplitudes and phases of the data, with further details as described previously in Methods: data reduction. Additional steps included calibration of the RCP–LCP phase and delay offsets, and determination of the D-term of each telescope. The broad bandwidth of 256 MHz makes it difficult to correct the delay offsets in the RL and LR polarisations using the task RLDLY. To deal with this issue, two 128-MHz intermediate frequency channels (IFs) were first divided into eight 32-MHz sub-IFs using the task MORIF. Then we run the CROSSPOL procedure to check and calibrate the RCP–LCP delay offsets. Additional details are presented in the NRAO AIPS memo 79 (ref. [41]). The AIPS task LPCAL was then used to determine the D-terms using the self-calibrated source model of 3C 84.

After the calibration of visibility phase, amplitude and instrumental polarisation, the J0906 + 6930 data were imported into the Caltech Difmap package for self-calibration and imaging. Several iterations of self-calibration and deconvolution were carried out until the signal-to-noise ratio in the image was below 5σ. After self-calibration, the Stokes I, Q and U components were separately imaged. The Stokes Q and U images were then combined to create the polarised intensity image using the AIPS task COMB, shown in Fig. 2. The rms noise in the combined polarised intensity image approaches the thermal noise 50 μJy beam$^{-1}$ estimated based on the observing time and telescope sensitivities. A 3σ threshold was used to remove the contamination from fake noise features when combining the polarised intensity image. In order to highlight the distribution of the polarised emission in the jet, the image was restored with a higher resolution beam, same as the 2017 image. The polarised emission peaks at 0.8 mas southwest of the core, aligned in the direction of J2 and is slightly eastward offset from J1. The fractional polarisation is ≈10%, peak intensity is ≈0.6 mJy beam$^{-1}$ (~12 times above the rms noise). This is the preliminary detection of linear polarisation from a z > 5 blazar, at a rest frame frequency ~100 GHz.

Although, as stated above, our project was not designed to reconstruct the EVPA of the yet to be detected polarised emission, once the detection was achieved, we reconstructed uncalibrated EVPA distribution following a standard procedure. EVPAs are calculated by using Stokes U and Q values, EVPA $= \frac{1}{2}\tan^{-1}\frac{U}{Q}$. Supplementary Fig. 2 represents the polarised emission image of J0906 + 6930 with uncalibrated distribution of EVPA. While specific orientation of the EVPA cannot be treated physically due to the absence of its calibration, the orderly distributed electric vectors are consistent with the shock-compressed magnetic field model.

**Radio light curve.** The source has been monitored by the Owens Valley Radio Observatory 40-m telescope monitoring programme[42] at 15 GHz from 2009. This enabled the verification of the amplitude calibration of the 15-GHz VLBA data. In Supplementary Fig. 1 we plot the light curve together with our VLBA measurements in the 2017 and 2018 epochs. The new VLBA measurements are consistent with the single-dish flux densities (the inset of Supplementary Fig. 1). This indicates that the integrated radio emission is dominated by the pc-scale compact core-jet. The dimming of the core and the brightening of J1 in 2017 and 2018 (in comparison to 2004 and 2005) possibly originates from the propagating shock (post the 2011 flare) interacting with the ISM.

**Doppler boosting parameters.** The brightness temperatures of the VLBI core

$$T_B = 1.22 \times 10^{12} \frac{S}{D_{maj}D_{min}\nu^2}(1+z) \text{ K,} \tag{1}$$

where S is the flux density (Jy) at the observing frequency ν (GHz), $D_{maj}$ and $D_{min}$ are the major and minor axis sizes of a Gaussian model (mas), and z is the redshift. $T_B$ ranges from 18.0 to 47.5 × 10$^{10}$ K with a mean $T_B$=(30.2 ± 4.0) × 10$^{10}$ K.

The inverse Compton catastrophe prevents the synchrotron brightness temperature from exceeding a threshold of ≈10$^{12}$ K[43]. An energy equipartition between magnetic fields and relativistic particles in a synchrotron radio source sets a lower maximum brightness temperature $T_{B,eq}$ of 5 × 10$^{10}$ K[32] than that theoretically expected[38,44]. $T_B$ in excess of $T_{B,eq}$ can be attributed to Doppler boosting of the relativistic jet beam. Using

$$T_B = \delta T_{B,eq}, \tag{2}$$

where $\delta \equiv [\Gamma(1 - \beta\cos\theta)]^{-1}$ is the Doppler factor, Γ is the bulk Lorentz factor and β is the jet bulk speed (in units of c), we obtain $\delta = 6.1 \pm 0.8$. The bulk Lorentz factor Γ and the viewing angle of the jet with respect to the observer LOS θ are

$$\Gamma = \frac{\beta_{app}^2 + \delta^2 + 1}{2\delta}, \tag{3}$$

$$\tan\theta = \frac{2\beta_{app}}{\beta_{app}^2 + \delta^2 - 1}. \tag{4}$$

With a jet apparent transverse speed $\beta_{app} = (2.5 \pm 0.8)c$ and $\delta = 6.1 \pm 0.9$, we obtain $\Gamma = 3.6 \pm 0.5$ and $\theta = 6.8° \pm 2.2°$. These values are consistent with the parametric fitting of the spectral energy distribution[27], where $\delta = 9^{+2.5}_{-3}$ and $\theta \leq 9.6°$ were obtained.

We also estimate the Doppler factor from the variability of radio flux density[45,46]. The monitoring data at 15 GHz from the OVRO 40-m telescope[42] is used for calculating the variability brightness temperature ($T_{\mathrm{B,var}}$). We modelled the flare with a Gaussian function:

$$S(t) = A \exp\left(-\frac{t - t_p}{2\tau^2}\right) + B,$$

where $S(t)$ is the source flux density (Jy), $t$ is the observation time (days), $B$ is the constant noise background (Jy), $t_p$ is the peak time (days) and $\tau$ is the characteristic flare rise timescale (days). We used a least-squares fit to estimate the Gaussian parameters, $t_p = 20$ May 2012, $\tau = 847$ days, $A = 0.149$ Jy, $B = 0.061$ Jy. The fitted parameters give a variability brightness temperature, $T_{\mathrm{B,var}} = 5.7 \times 10^{12}$ K. The corresponding Doppler factor is $\delta_{\mathrm{var}} = \sqrt[3]{\left(\frac{T_{\mathrm{B,var}}}{T_{\mathrm{B,eq}}}\right)} = 4.8$. This value is consistent but marginally lower than that derived from the above VLBI model fitting.

**Maximum proper motion**. The proper motion can be expressed as

$$\mu = \frac{\beta_\perp c a}{D_A},$$

where $\beta_\perp$ is the transverse jet speed, $a = (1 + z)^{-1}$ and $D_A$ is the angular size distance. This can be approximated in terms of the Hubble distance $c/H_0$ as[47]

$$D_A = \frac{c}{H_0} \frac{a}{g(a)},$$

where

$$g(a) = \left(\frac{a}{1-a} + 0.2278 + \frac{0.027(1-a)}{0.785 + a} - \frac{0.0158(1-a)}{(0.312 + a)^2}\right).$$

Using Eqs. (6), (7) and $\beta_\perp \approx \Gamma\beta$ (for an extremely beamed jet) in Eq. (5), the maximal proper motion

$$\mu_{\max} \le H_0 \Gamma\beta g(a)^{-1}$$
$$\mu_{\max} = (0.1\ \mathrm{mas\ yr^{-1}}) \left(\frac{\Gamma}{3.6}\right) \left(\frac{H_0}{70\ \mathrm{km\ s^{-1}\ Mpc^{-1}}}\right)$$

for $z = 5.47$ and $\beta = 1$.

**Black hole mass and jet activity**. The previous black hole mass estimate of $M_{\mathrm{BH}} = 4.2 \times 10^9 M_\odot$[27] is in tension with the inferred observational signatures of the source resembling a high-frequency peaker, which typically have moderately lower masses in the range $10^{7.5}$–$10^8 M_\odot$ ref. [48]. The earlier estimates are based on scaling relations relevant to non-jetted quasars. However, the employed luminosity requires to be Doppler beaming corrected for a jetted quasar[49]. For the blazar J0906 + 6930, comparing the scaled broad line region (BLR) radius

$$r_{\mathrm{BLR}} = (5.8 \times 10^{16}\ \mathrm{cm}) \left[\frac{\lambda L_\lambda (1350\text{Å})}{10^{44}\mathrm{erg\,s^{-1}}}\right]^{0.61}$$ ref. [49] with $r_{\mathrm{BLR}} = (9.54 \times 10^{16}\ \mathrm{cm})(\dot{m}\,m_9)^{0.5}$

where $\dot{m}$ is the mass accretion rate scaled in Eddington units and $m_9 = \frac{M_{\mathrm{BH}}}{10^9 M_\odot}$ ref. [50] results in $(\lambda L_\lambda(1350\ \text{Å})) = (2.6 \times 10^{44}\ \mathrm{erg\,s^{-1}})(\dot{m}\,m_9)^{0.82}$. Using this in the revised scaling relation $m_9 = 3.26 \times 10^{-3} \left[\frac{\lambda L_\lambda(1350\text{Å})}{10^{44}\mathrm{ergs^{-1}}}\right]^{0.61} \left[\frac{v}{10^3 \mathrm{km\,s^{-1}}}\right]^2$ where $v$ is the full-width at the half-maximum velocity as inferred from the emission line[49], for $\dot{m} \le 1$ (ref. [51]) and $v = 6000$ km s$^{-1}$ (ref. [27]), we obtain $M_{\mathrm{BH}} = 4.4 \times 10^7\ M_\odot$ which is consistent with the above expectation from the population of sources. The lower mass and a similarity to a young AGN indicate the ongoing evolution of the black hole[48], with feedback transitioning from accretion dominated (radiative) to the onset of the jet (momentum) driving possibly resulting in a black hole–galaxy co-evolution.

The formation and rapid growth of SMBHs in the early Universe remains debatable. That can be enabled by a combination of accretion and fuelling, AGN feedback through radiation and jet interaction with the large-scale environment, and galaxy mergers. The accompanying jet activity includes a component ejection associated with a major flare with nearly constant amplitudes and lack of a time delay when observed at different frequencies[52]. The jet structure and kinematic evolution can result from scenarios alternative to the bending due to interaction. These include a helical trajectory[53], instabilities in the jet[54,55] possibly induced by the surrounding environment or through turbulent loading at the jet base[56,57]. In these cases, the jet is non-nascent and could either be an ongoing process or episodic. This would indicate a departure from the young AGN scenario, and point towards a relatively lower powered evolved blazar with possibly large-scale morphology. These can be addressed through the continued observation of this and similar sources at various physical scales (pc~kpc) and from an accumulated statistically viable sample.

**Jet and ISM properties**. Assuming the convention $S_\nu \propto \nu^\alpha$, a core spectral index of $\alpha = 0$ is assumed based on the inference of a spectral turnover near 10 GHz[15,16], also indicated by similar core flux densities at 8.4 and 15 GHz during the 2004

epoch[15]. The average monochromatic radio power of the core-jet,

$$\bar{P}_\nu = 4\pi D_L^2 \bar{S}_\nu (1 + z)^{-1-\alpha}.$$

With a luminosity distance $D_L = 53.66$ Gpc ($z = 5.47$ for J0906 + 6930) obtained using CosmoCalc[58], $\bar{S}_{\mathrm{core,15GHz}} = 90.94 \pm 5.18$ mJy and the above $\alpha = 0$, $L_{\mathrm{radio}} \approx \nu_{15\,\mathrm{GHz}} \bar{P}_{\mathrm{core,15GHz}} = (7.28 \pm 0.42) \times 10^{44}$ erg s$^{-1}$. A standard ΛCDM cosmological model with $H_0 = 70$ km s$^{-1}$ Mpc, $\Omega_\Lambda = 0.73$ and $\Omega_M = 0.27$ was used. The radiative and kinetic contributions to the jet luminosity $L_{\mathrm{jet}}$ are related to the radio luminosity by the empirical relations[59]

$$\log L_{\mathrm{jet,rad}} = (12 \pm 2) + (0.75 \pm 0.04) \log L_{\mathrm{radio}},$$

$$\log L_{\mathrm{jet,kinetic}} = (6 \pm 2) + (0.90 \pm 0.04) \log L_{\mathrm{radio}}$$

The jet luminosity is taken as the addition of the radiative and kinetic contributions and is $L_{\mathrm{jet}} = 2.8 \times 10^{46}$ erg s$^{-1}$, and is $2.2 \times 10^{43}$ erg s$^{-1}$ using $\delta = 6.0$ as inferred in Appendix F. This is ~0.02 $L_{\mathrm{Edd}}$, where $L_{\mathrm{Edd}} = (1.3 \times 10^{47}$ erg s$^{-1})m_9$ is the Eddington luminosity and using $m_9 = 4.4 \times 10^{-2}$ as inferred from Appendix H. It must be noted that the empirical relations have a large scatter mainly due to the diversity in the sources constituting the inference, ranging from radio-loud AGN to X-ray binaries, and having to properly account for spectral state transitions, variability and systematics from comparison of data from different databases[59]. The uncertainty on the jet power may then be underestimated.

Under conditions of an ambient ISM (non-relativistic), the momentum flux balance between the jet and the ISM[60]

$$\frac{L_{\mathrm{jet}}}{(\beta_{\mathrm{jet}} - \beta_{\mathrm{h}})c} = \rho_{\mathrm{ISM}} \beta_{\mathrm{h}}^2 c^2 A_{\mathrm{h}},$$

where $L_{\mathrm{jet}}$ is the jet luminosity, $\beta_{\mathrm{jet}}$ is the jet speed (in units of $c$), $\rho_{\mathrm{ISM}}$ is the ISM density, $\beta_{\mathrm{h}}$ is the speed of the advancing jet head (in units of $c$) and $A_{\mathrm{h}}$ is the surface area of the jet head interacting with the ISM.

The average core separation from J1, $r_{\mathrm{J1}}/\sin\theta = 43.2$ pc from which, the circumferential radius in contact with the ISM $r_{\mathrm{h}} = r_{\mathrm{J1}}/2 = 21.6$ pc and $A_{\mathrm{h}} \approx r_{\mathrm{h}}^2 \sin\theta_0 = 82.9$ pc$^2$ assuming an intrinsic jet half-opening angle $\theta_0 = 10°$ ref. [61]. The relativistic transformation,

$$\beta = \frac{\beta_\perp}{\sin\theta + \beta_\perp \cos\theta}$$

relates the projected transverse speed (proper motion) $\beta_\perp$ with the true speed $\beta$ through the inclination angle $\theta$. The jet speed $\beta_{\mathrm{jet}} \sim \beta_{\mathrm{J2}} = 0.96\,c$ based on a $\beta_{\perp,\mathrm{J2}} = 2.5\,c$ and using the above relativistic transformation. The speed of the jet head $\beta_{\mathrm{h}} \le \beta_{\mathrm{J2}}/4 = 0.24\,c$ (upper limit on the speed of the shocked expanding gas at the jet head post interaction with the ISM), assuming a strong shock interaction which entails a post shock plasma with the above maximal speed. The density contrast between the ISM and jet[60] is

$$\frac{\rho_{\mathrm{ISM}}}{\rho_{\mathrm{jet}}} = \left(\frac{\beta_{\mathrm{jet}}}{\beta_{\mathrm{h}}} - 1\right)^2$$

using which $\rho_{\mathrm{ISM}}/\rho_{\mathrm{jet}} > 9$, likely resulting in the lower density jet being susceptible to a deflection or bending upon interaction with the ISM, while being able to cause a compression of the material near the jet head. Employing the above calculated quantities and the simplifying assumptions in Eq. (12), $\rho_{\mathrm{ISM}} \ge 2.4 \times 10^{-26}$ g cm$^{-3}$ which corresponds to a number density $n_e \ge 26.6$ cm$^{-3}$.

## Data availability

All data used in this study are public and can be accessed through the different data archives of the various instruments. NRAO VLBA archive: https://archive.nrao.edu/archive/advquery.jsp, OVRO archive: http://www.astro.caltech.edu/ovroblazars/data.php?page = data_query. The authors can provide data supporting this study upon request.

## Code availability

Upon reasonable request the authors will provide all code supporting this study. Astronomical Image Processing System (AIPS) software can be found at http://www.aips.nrao.edu/index.shtml. Difmap software can be found at ftp://ftp.astro.caltech.edu/pub/difmap/.

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

## Acknowledgements

This work is funded by the National Key R&D Programme of China (2018YFA0404603), the Chinese Academy of Sciences (CAS, 114231KYSB20170003), and the Hungarian National Research, Development and Innovation Office (2018-2.1.14-TET-CN-2018-00001). P.M. is supported by CAS-PIFI (2016PM024) post-doctoral fellowship and the NSFC grant (11650110438). K.É.G. was supported by the János Bolyai Research Scholarship of the Hungarian Academy of Sciences and by the ÚNKP-19-4 New National Excellence Program of the Ministry for Innovation and Technology. This research has made use of data observed with the Very Long Baseline Array of the National Radio Astronomy Observatory (project codes: BR093, BG154, BZ068, BZ071). The NRAO is a facility of the National Science Foundation operated under cooperative agreement by Associated Universities, Inc. We acknowledge the use of calibrated visibility data from the Astrogeo Center Database maintained by L. Petrov and the light curve data from the OVRO 40-m monitoring program which is supported in part by NASA grants NNX08AW31G, NNX11A043G, and NNX14AQ89G and NSF grants AST-0808050 and AST-1109911.

## Author contribution

T.A. and P.M. wrote the initial manuscript. T.A. and Y.Z. led the VLBA observations. S.F. and J.Y. contributed to the design and implementation of the observations. K.E.G., Z.P., L.I.G., K.P. and Z.Z. contributed to data analysis. All co-authors read and contributed to the manuscript and supplementary information.

## Competing interests

The authors declare no competing interests.
