## [Peer Review File · Nature Communications]

Reviewers' comments:

Reviewer #1 (Remarks to the Author):

Referee's Report for Tao An et al MS 213259

This paper discusses some new VLBI measurements of the most distant blazar, J0906+6930. The VLBI maps are nice and make an improvement over the previous publication in Zhang et al 2017. The new detection of significant (10\%) linear polarization is what is featured here. The paper proceeds to some interpretation of component motion in the multi-epoch VLBI maps.

On the whole the paper is fine, with a useful addendum of information on an interesting source. I found it well worth reading. A few aspects did bother me though.

1) It is rather disturbing to see in Figure 2 the relatively strong linear polarization flux with no indication of EVPA. I see that in Appendix D the authors note that they did not attempt to get the EVPA; why not?? You made the Q and U maps; if you trust these we should have a EVPA value, if not, why do you trust your P_i ? The leakage terms should be known, albeit perhaps imperfectly.

The EVPA is the most important datum for physical interpretation; this is the aspect that is likely to influence scientists in this field. Thus, I believe that you should plot an EVPA estimate on Figure 2 (with all appropriate caveats), and discuss at least briefly.

If the data do not allow any useful estimate, please state clearly why and why the Π measurement is then still robust.

2) The interpretation focuses on the idea that J1 is a termination shock with the deflected jet represented by J2, motivated by the apparent stable core-J1 distance. I think this is plausible, but surely there are other interpretations -- perhaps J1 and the core are two standing shocks in the outflow. So while the idea that this is a nascent jet tunneling through the host ISM is interesting, some alternatives should be mentioned. One motivation for doing this is that the present apparently substantially sub-Eddington jet luminosity may not be a characteristic state for such a large BH mass at high z . One might expect an earlier, more typical, high luminosity phase to have cleared the local ISM. Perhaps the BH has recently entered the present dense, uncleared ISM, but this seems odd.

3) Related to 2): looking through the appendices it appears that the authors have been somewhat cavalier in their model estimates. Given that these are intrinsically imprecise the general conclusions may be OK, but the logic seems a bit forced. For example, they use the $\beta_{\text{perp}} \sim 2.2$ of J2 in a number of equations when evaluating the properties of the shock at J1. For example the $\theta \sim 7^\circ$ is applied to J1 to get a true core separation. These numbers are also used in the penultimate paragraph of the main article, which further obscures that this β_{perp} is being applied to the pre-J1 flow. (Also the argument around Eq 13 is circular -- β_{h} was just inferred for a strong shock, which assumes the high density contrast.) These examples suggest that, at

a minimum the authors need to be clearer about what they have assumed in making their estimates (e.g. small J1/J2 angle deflection, no deceleration).

The authors did argue that the 2017 J1 flux was responding to the 2011 core flare. That could give an effective β_{\perp} estimate for pre-J1 flow of $17.3 \text{ lt-yr} / 6 \text{ yr} > 2.9$. That is crude but it does apply to J1.

4) Still in Appendix H, I was somewhat puzzled why the authors used the monochromatic expression from Birzan et al. (Eq 16) in Eq 10. Since one sees an SED peak in the radio we have a \sim bolometric luminosity so the bolometric expression from Birzan (Eq 13) would seem more appropriate; that gives $L_j \sim 3.4 \times 10^{45} \text{ erg/s}$ and $M_{\text{edd}} \sim 0.03$ for $10^9 M_{\text{sun}}$, with the numbers quoted. A more careful integration of the rest frame radio luminosity should be used, though. On the other hand An et al (ref 25) give M as $4 \times 10^9 M_{\text{sun}}$, which re-lowers the efficiency -- you might wish to use that mass in your estimates or at least mention it and give the scaling. On the subject of the radio SED peak -- this was published in Romani 2006 AJ 132, 1959. the reference used (14 = Zhang et al 2017) largely drew the data from that paper and I don't think it materially improved the spectrum measurement on that earlier result.

The Coppejans reference in App. H should be 2016, not 2017.

Overall I think that this is a good paper, but it could be better. If, as the authors repeatedly note, measuring the polarization is a big deal,

they should give an EVPA estimate. If the nascent jet/ISM shock argument is the most important result, the arguments about its energetics should be clearer.

Perhaps with a bit of attention to the points above these aspects could be improved.

Publication would then be appropriate.

Reviewer #2 (Remarks to the Author):

The paper reports on multi-epoch VLBI observations of the highest-redshift ($z=5.47$) blazar J0906+6930 to measure proper motion and polarization of its jet. Using the astrometry and polarimetry observations, the authors found that the bulk Lorentz factor of the jet is >3.3 with a viewing angle of <7.1 degree, and the jet luminosity is 8.2×10^{44} erg/s (6×10^{-3} Eddington luminosity), suggesting that the blazar's jet is relatively less powerful (compared to those in nearby Universe). In addition, the authors infer an ISM density of 65 cm^{-3} , >11 times denser than the jet. This together with the jet bending implies that the low-density jet is interacting with dense ambient medium. These results from the first proper-motion and polarization measurement of the record-holding blazar is certainly very interesting and can help researchers in the field to further understanding of high- z blazars and early Universe.

The paper is well organized, but I do not recommend that this paper is accepted for publication in Nature Communications for reasons shown below.

While observations of record-holding sources often provide new insights into physics in similar sources, it is not clearly presented in the manuscript that their results do so. The radio polarization measurement for such a high- z blazar is new, but the measurement seems not to add significantly new information to the results (other than strengthen the argument). It seems that the other measurements (e.g., proper motion) also mainly strengthen previous suggestions/conclusions for the same source and similar high- z blazars, and I do not find their interpretations and conclusions are novel. One can find similar interpretations/conclusions in (or directly infer from) previous works (e.g., a less powerful jet of J0906 in Zhang et al. 2017, An and Romani 2018, and jet bending due to interaction with ambient medium for J0906 or other high- z blazars in Frey et al. 2014, Frey et al. 2018). Hence, the paper seems not to significantly influence current thinking of high- z blazars or early Universe.

For the analyses, the authors used standard techniques (e.g., using the brightness temperature calculation to derive the bulk Lorentz factor, and an empirical relation for jet-power calculation) and

assumptions. These are often accepted in the community and used in literatures, and I do not find that the techniques/assumptions are problematic. However, they make the authors' conclusions suggestive rather than definitive. That is, the results seem not to support strongly the conclusions due to assumptions used, uncertainties in the measurements and/or lack of justification for assumptions/calculations (see below for more details).

Below are my suggestions to improve the paper. Addressing the issues may help to improve the paper but may not be sufficient to make the paper acceptable.

* Comments:

These are mainly about handling uncertainties when the authors calculate derived quantities from measured ones.

(1) Confidence intervals for the errors need to be given: 68% or 90% or something else? If 68%, the proper motion measurement may not be significant enough; they are only at the ~ 2 -sigma level (95%). In this case, reporting upper limits may be more appropriate.

(2) In Appendix C, it is said that the positional errors are 10% and 5% of FWHM of the restoring beam for J1 and J2, respectively. As the errors may have significant impacts on the (proper motion and other) results, these numbers need further justifications.

(3) The authors do not propagate measurement errors adequately when reporting other derived quantities. Some of their measurements have relatively large uncertainties (e.g., proper motion), and so their conclusions may change if considering the errors. It may be very difficult to quantify all the errors, but they need to be considered seriously when drawing conclusions.

(4) The bulk Lorentz factor is reported to be >3.3 , only an upper limit. While this is consistent with previous estimations (e.g., Zhang et al. 2017, An & Romani 2018), it alone does not rule out a possibility of a powerful jet (as opposed to the claimed 'less powerful jet').

(5) For inferring the jet power and ISM density (Appendix H), the authors assume that the spectral index is 0 between 1.4 GHz and 15 GHz for J1 and/or J2. Although previous measurements of the radio spectrum seem to support this albeit a spectral break at ~ 10 GHz, the measurements are for C+J1+J2 (or C as it dominates) as compared to J1+J2. Is it reasonable to assume the same spectrum for J1+J2 when the variability of C+J1+J2 is different from that of J1 (and/or J2) in Table 3 and Figure

A1? This may be a concern when extrapolating the 15-GHz flux density to 1.4-GHz. In particular, the authors mention a propagating shock. The propagating shock may change the radio spectral shape in time (e.g., Valtaoja et al. 1992), hence assuming a radio spectral index of 0 in 2017-2018 based on earlier data needs a justification. In this regard, (near) contemporaneous multi-frequency observations may be helpful.

(6) The empirical relation (between L_{jet} and $P_{1.4\text{GHz}}$) they used in Appendix H may have a relatively large uncertainty (a few orders of magnitude scatter in the data used for deriving the relation; Birzan et al. 2008); this (and other) uncertainty needs to be considered when deriving the jet power and claiming that it is low.

(7) The reported number $n \sim 65 \text{ cm}^{-3}$ (in the text and Appendix H) for the ISM density is too precise compared to the arguments given in Appendix H (see also comments 3-5). Shouldn't it be a limit as is for the density contrast $\rho_{\text{ISM}}/\rho_{\text{jet}} > 11$ because β_h is only a limit?

(8) It is said "In the highest-resolution images, a sharp (>90 degree) jet bending is seen..." in the text. As the jet bending is seen in a previous study of the same source (e.g., Zhang et al. 2017), the previous work needs to be recognized.

(9) It is said "a prominent X-ray emission (with a strong contribution from the disk).." in the text. It is believed that X-rays in these high- z blazars are produced mainly by the synchrotron-self-Compton process and that disks of supermassive blackholes mostly radiate in the optical band. Hence, the statement needs to be clarified.

(10) Contours in Figure 1-e and Figure 2 are slightly different although it appears that they are generated from the same observational data.

(11) In Appendix H, it appears that the authors consider J_1+J_2 . This should be mentioned explicitly. Assuming J_1+J_2 , I do not reproduce $S_{15\text{GHz}}=21 \text{ mJy}$ from the table. In 2017-2018, the average flux of J_1 and J_2 is $\sim 20.4 \text{ mJy}$. The difference is only tiny, but I wonder if there is something else I am missing here.

(12) In Appendix G above equation (8) it is said "Employing equation (7) which reduces to...". Equation (7) does not seem to reduce to (8), so this may be a mistake.

Reviewer #3 (Remarks to the Author):

The essay deals with the changes in the jet structure from high-resolutions VLBI radio observations of the jetted AGN J0906+6930 ($z=5.47$). Authors find changes in the jet direction and interpret them as the interaction with a dense interstellar medium, suggesting a young age for this AGN. This topic is important and could be of interest for a larger public, as it addresses the presence of supermassive black holes (SMBH) in a relatively young universe. Since there is no time enough to form a SMBH via a sequence of mergers of smaller black holes, the detection of SMBH in the early universe sets the question of how they are formed. The essay could be published, but after some points that must be addressed by the authors. Specific comments follows:

- Authors indicated J0906+6930 as the most distant known blazar, having $z=5.47$. However, there is a recent detection by Saxena et al. (2018, MNRAS, 480, 2733) of a jetted AGN at $z=5.72$. The radio spectral index suggests it could be a radio galaxy, thus with the jet viewing angle greater than $\approx 10^\circ$. Therefore, the authors statements are formally correct. However, I would say a few words about this other jetted AGN.

- page 1, row 19: Authors wrote: "As jetted quasars seem to harbour heavier black holes than non jetted ones, high-redshift blazars are probes...". The line of reasoning is inverted. High- z quasars are useful probes for the early universe because they are the brightest sources, and, therefore, it is easier to find them. But, the extreme brightness is due to: (1) the presence of a relativistic jet viewed by a small angle, which means that the electromagnetic emission is Doppler boosted (with a typical Doppler factor of 10, the luminosity of the jet is boosted to 10000 times its isotropic value); (2) as the jet power scales with the mass of the central black hole (P proportional to $M^{1.4}$, cf Heinz & Sunyaev 2003), a larger mass implies a larger observed jet power.

- page 1, row 28-29: the radio loudness is a meaningless parameter. No need to cite it.

- Section F: Doppler boosting parameter of the jet. Authors estimated the Doppler factor δ by using the brightness temperature as measured from the VLBI core size. The apparent jet speed β_{app} is measured from VLBI observations. From δ and β_{app} , it is then possible to calculate the bulk Lorentz factor Γ and the viewing angle θ . As the authors reported also the OVRO 15 GHz radio light curve, I suggest to calculate δ also from radio variability. An easier way is to refer to Lioudakis et al. (2018, ApJ, 866, 137), who already did it and reported $\delta=8.91$. Together with $\beta_{\text{app}}=2.2c$ measured by the authors, it results $\Gamma=4.8$ and $\theta=3^\circ$. Please note that

Liodakis calculated the equipartition brightness temperature is a way different from usual. So, when comparing your results with him, please take into account this difference. Please note that also the main text -- not only Section F -- must be changed accordingly.

- Section H: authors calculated the jet power by using a relationship by Birzan et al. (2008). These relationships links the extended radio emission measured in the MHz frequency range with the jet kinetic power. Therefore, the application to one component detected at 15 GHz is rather risky, as it is known that extrapolating from high to low frequency could give serious errors in the estimation of the extended emission. In addition, the jet power by Birzan refers only to the kinetic component, and does not include the radiative part. The kinetic part calculated in this way is based on the cavities excavated by the jet activity during million of years and are not comparable with a year time scale as from the authors of the present essay.

I suggest to adopt the relationships by Foschini (2014, Int. J. Mod. Phys. Conf. Series, 28, 1460188), which links the radiative and kinetic power to the 15 GHz core luminosity. You should apply them to the core flux, but you can also try to apply them to the component J1. Obviously, you have to address the proper caveats. A quick calculation by using the core emission, resulted in a $\log P_{\text{rad}}=45.4$ [erg/s] and $\log P_{\text{kin}}=46.1$ [erg/s], greater than the estimates from the Birzan relationship, as expected.

- the last (but not least) point refers to the estimation of the mass of the central black hole and the accretion rate. Authors refer to previous works, which gave a mass of the order of $10^{9.9} M_{\text{Sun}}$ and accretion luminosity 0.4 times the Eddington limit. However, all the previous estimates adopted relationships made for non-jetted AGN. The presence of the Doppler boosting alters significantly the continuum emission, which implies an overestimation of the mass. Also the fit of the accretion disk suffers the same problem. As the optical spectra all show a strong Ly α emission line, I suggest to use it together the relationships by Pian et al. (2005, MNRAS, 361, 919), which take into account the Doppler boosting. Indeed, a quick calculation of the $\lambda L(1350\text{\AA})$ by using the Pian's relationships results in a value of $\approx 7.4 \times 10^{43}$ erg/s to be compared with the value of 5×10^{47} erg/s given by Romani (2004). The radius of the broad-line region (BLR) from Bentz et al. (2013), results to be 18.6 light days, which implies a luminosity of the disk of $\approx 3 \times 10^{43}$ erg/s. I've not found a measurement of the FWHM of Ly α , but if we assume the value of 5000 km/s from other lines, the mass of the central black hole results to be $\approx 7 \times 10^7 M_{\text{Sun}}$.

Therefore, I suggest to recalculate the mass and accretion rate of the central black hole by using the Ly α line parameters and Pian's relationships. Then, revise the discussion in the text according to the new values. A smaller mass of the central black hole would be more suitable for a young AGN, and also more consistent with a high-frequency peaker (HFP). Compare also with Berton et al. (2016, A&A, 591, A98).

Dear Referees,

We are very grateful for the detailed reviews and constructive comments and suggestions on our manuscript. We re-worked its contents attempting to address all concerns expressed in the reviews. Below we address each point. The review items are shown in blue, slanted and indented fonts and our replies are in straight black fonts.

Tao An
On behalf of co-authors

=====

Reviewer #1 (Remarks to the Author):

Referee's Report for Tao An et al MS 213259

This paper discusses some new VLBI measurements of the most distant blazar, J0906+6930. The VLBI maps are nice and make an improvement over the previous publication in Zhang et al 2017. The new detection of significant (10%) linear polarization is what is featured here. The paper proceeds to some interpretation of component motion in the multi-epoch VLBI maps.

On the whole the paper is fine, with a useful addendum of information on an interesting source. I found it well worth reading. A few aspects did bother me though.

1). It is rather disturbing to see in Figure 2 the relatively strong linear polarization flux with no indication of EVPA. I see that in Appendix D the authors note that they did not attempt to get the EVPA; why not?? You made the Q and U maps; if you trust these we should have a EVPA value, if not, why do you trust your Pi? The leakage terms should be known, albeit perhaps imperfectly. The EVPA is the most important datum for physical interpretation; this is the aspect that is likely to influence scientists in this field. Thus, I believe that you should plot an EVPA estimate on Figure 2 (with all appropriate caveats), and discuss at least briefly. If the data do not allow any useful estimate, please state clearly why and why the Pi measurement is then still robust.

The observing project presented in this paper was designed as an exploratory one aiming to detect the polarised emission at such a high redshift. The total on-target time was limited to 2 hours, not allowing us to conduct a proper EVPA calibration procedure (the latter requires a sizeable fraction of the overall experiment duration to spend on a calibrator source with known EVPA properties).

Nevertheless, after successful detection of the compact polarised emission in the target source, we conducted a formal procedure of reconstructing the EVPA distribution using the Stokes U and Q values, $EVPA = \frac{1}{2} \tan^{-1} \frac{U}{Q}$. Fig. A1 added to the revised Appendix D of the new version of the manuscript shows this distribution, clearly stating that these are uncalibrated data. However, the orderly character of the uncalibrated EVPAs evident in this image is consistent with the shock-compressed magnetic field model.

Therefore, the revised Appendix D addresses this concern of the reviewer #1 by addition of several sentences of paragraph 1 and new figure with corresponding explanation at the end of the Appendix.

2) The interpretation focuses on the idea that J1 is a termination shock with the deflected jet represented by J2, motivated by the apparent stable core-J1 distance. I think this is plausible, but surely there are other interpretations -- perhaps J1 and the core are two standing shocks in the outflow. So while the idea that this is a nascent jet tunneling through the host ISM is interesting, some alternatives should be mentioned. One motivation for doing this is that the present apparently substantially sub-Eddington jet luminosity may not be a characteristic state for such a large BH mass at high z. One might expect an earlier, more typical, high luminosity phase to have cleared the local ISM. Perhaps the BH has recently entered the present dense, uncleared ISM, but this seems odd.

The resolution achieved in our experiment does not allow to resolve the core. The observed emission is likely to be a blend of the synchrotron self-absorbed core (at the observing frequency of 15 GHz) and inner jet. Both J1 and the core could be standing shocks in the jet flow. We have added a sentence to clarify this in the context of the increased jet emission in the 7th paragraph of the manuscript.

An alternative to the nascent jet scenario, a re-started jet along a swept path by a past jet is also possible. Verification of such the possibility would require an intermediate-resolution images of the extended emission structure on the scale of 10^2 mas. This is beyond the scope of the project presented here. Nevertheless, three sentences addressing this alternative source model are added at the end of the 7th paragraph of the main text.

3) Related to 2): looking through the appendices it appears that the authors have been somewhat cavalier in their model estimates. Given that these are intrinsically imprecise the general conclusions may be OK, but the logic seems a bit forced. For example, they use the $\beta_{\text{perp}} \sim 2.2$ of J2 in a number of equations when evaluating the properties of the shock at J1. For example the $\theta \sim 7$ deg is applied to J1 to get a true core separation. These numbers are also used in the penultimate paragraph of the main article, which further obscures that this β_{perp} is being applied to the pre-J1 flow. (Also the argument around Eq 13 is circular -- β_{h} was just inferred for a strong shock, which assumes the high density contrast.) These examples suggest that, at a minimum the authors need to be clearer about what they have assumed in making their estimates (e.g. small J1/J2 angle deflection, no deceleration).

The authors did argue that the 2017 J1 flux was responding to the 2011 core flare. That could give an effective β_{perp} estimate for pre-J1 flow of $17.3 \text{lt-yr} / <6y > 2.9$. That is crude but it does apply to J1.

The component J1 is likely to mark the site where the jet beam head encounters a dense medium. A jitter of the jet head at the jet-ISM interface results in a random position change of the hotspot J1. It might also result in the non-detection of the J1 proper motion along the core-jet direction. In contrast, the outer jet component J2 has a significant proper motion, indicating the expansion of the jet structure. In order to estimate the jet parameters, we have used a simplified but reasonable assumption in which the bulk speed of the jet flow is constant in the pre-J1 and post-J1 sections. The variability of the core and J1 and the time delay between them supplements this notion.

Improved constraints on jet apparent transverse speed β_{app} are now used in a revised Appendix I, also addressing the above concerns on the density contrast.

4) Still in Appendix H, I was somewhat puzzled why the authors used the monochromatic expression from Birzan et al. (Eq 16) in Eq 10. Since one sees an SED peak in the radio we have a \sim bolometric luminosity so the bolometric

expression from Birzan (Eq 13) would seem more appropriate; that gives $L_j \sim 3.4e45$ erg/s and $M_{\text{edd}} \sim 0.03$ for $10^9 M_{\text{sun}}$, with the numbers quoted. A more careful integration of the rest frame radio luminosity should be used, though. On the other hand An et al (ref 25) give M as $4 \times 10^9 M_{\text{sun}}$, which re-lowers the efficiency -- you might wish to use that mass in your estimates or at least mention it and give the scaling.

New estimates are made using the jet power - radio luminosity relationship presented in Foschini (2014) in Appendix I.

On the subject of the radio SED peak -- this was published in Romani 2006 AJ 132, 1959. the reference used (14 = Zhang et al 2017) largely drew the data from that paper and I don't think it materially improved the spectrum measurement on that earlier result.

The SED reference Ref.15 (Romani 2006) is added in mid of 7th paragraph of the main text and in Appendix I. Other consequent reference numbers are corrected accordingly.

The Coppejans reference in App. H should be 2016, not 2017.

Corrected in reference #5.

Overall I think that this is a good paper, but it could be better. If, as the authors repeatedly note, measuring the polarization is a big deal, they should give an EVPA estimate. If the nascent jet/ISM shock argument is the most important result, the arguments about its energetics should clearer. Perhaps with a bit of attention to the points above these aspects could be improved. Publication would then be appropriate.

The EVPA calibration, nascent jet argument and other comments are answered above and the EVPA issue is now addressed in the new version of Appendix D.

Reviewer #2 (Remarks to the Author):

The paper reports on multi-epoch VLBI observations of the highest-redshift ($z=5.47$) blazar J0906+6930 to measure proper motion and polarization of its jet. Using the astrometry and polarimetry observations, the authors found that the bulk Lorentz factor of the jet is >3.3 with a viewing angle of <7.1 degree, and the jet luminosity is $8.2e44$ erg/s ($6e-3$ Eddington luminosity), suggesting that the blazar's jet is relatively less powerful (compared to those in nearby Universe). In addition, the authors infer an ISM density of 65 cm^{-3} , >11 times denser than the jet. This together with the jet bending implies that the low-density jet is interacting with dense ambient medium. These results from the first proper-motion and polarization measurement of the record-holding blazar is certainly very interesting and can help researchers in the field to further understanding of high- z blazars and early Universe.

The paper is well organized, but I do not recommend that this paper is accepted for publication in Nature Communications for reasons shown below.

While observations of record-holding sources often provide new insights into physics in similar sources, it is not clearly presented in the manuscript that their results do so. The radio polarization measurement for such a high- z blazar is new, but the measurement seems not to add significantly new information to the results (other than strengthen the argument). It seems that the other measurements (e.g., proper motion) also mainly strengthen previous suggestions/conclusions for the same

source and similar high-z blazars, and I do not find their interpretations and conclusions are novel. One can find similar interpretations/conclusions in (or directly infer from) previous works (e.g., a less powerful jet of J0906 in Zhang et al. 2017, An and Romani 2018, and jet bending due to interaction with ambient medium for J0906 or other high-z blazars in Frey et al. 2014, Frey et al. 2018). Hence, the paper seems not to significantly influence current thinking of high-z blazars or early Universe.

For the analyses, the authors used standard techniques (e.g., using the brightness temperature calculation to derive the bulk Lorentz factor, and an empirical relation for jet-power calculation) and assumptions. These are often accepted in the community and used in literatures, and I do not find that the techniques/assumptions are problematic. However, they make the authors' conclusions suggestive rather than definitive. That is, the results seem not to support strongly the conclusions due to assumptions used, uncertainties in the measurements and/or lack of justification for assumptions/calculations (see below for more details).

Below are my suggestions to improve the paper. Addressing the issues may help to improve the paper but may not be sufficient to make the paper acceptable.

* Comments: These are mainly about handling uncertainties when the authors calculate derived quantities from measured ones.

(1) Confidence intervals for the errors need to be given: 68% or 90% or (something else? If 68%, the proper motion measurement may not be significant (enough; they are only at the ~2-sigma level (95%). In this case, reporting (upper limits may be more appropriate. 2) In Appendix C, it is said that the (positional errors are 10% and 5% of FWHM of the restoring beam for J1 and J2, (respectively. As the errors may have significant impacts on the (proper motion (and other) results, these numbers need further justifications.

In the original version of the manuscript, we used experiential methods to estimate the measurement uncertainty. For example, 10% beam size was used as a positional error. Following the reviewer's remarks, we have carefully re-analyzed the uncertainty of the fitted models.

The statistical errors from the fitted Gaussian models are rough estimates based on the signal-to-noise levels in the images, introduced in Fomalont (1999). In addition to the statistical error $\sigma_{S,sta}$, the true errors can comprise contributions from other factors resulting from the calibration error of the flux density scale, incomplete u,v -coverage of the interferometric array and the complex jet structure. An extra 5 per cent of the flux density calibration error, which is the typical value used for the VLBA (Lister et al. 2016), is added to account for the calibration error $\sigma_{S,cal}$ of the visibility amplitude. Thus, the total uncertainty of

the flux density is estimated as:
$$\sigma_S = \sqrt{\sigma_{S,cal}^2 + \sigma_{S,sta}^2} .$$

The uncertainty of the Gaussian component size is decomposed into the statistical error and the fitting error. The final size uncertainty is the quadratic mean of these two errors.

The statistical error is calculated using the Fomalont (1999) equation : $\sigma_{R,sys} = \theta_D / \text{SNR}$, where the SNR is the signal-to-noise ratio of the fitted Gaussian component, θ_D is the fitted Gaussian size with the corresponding D_{maj} and D_{min} listed in columns 4 and 5 of Table A3.

The positions of jet components are measured with respect to the core which is assumed to be stationary. The statistical error of the position is given as $\sigma_{R,sta} = \frac{\theta}{2 \cdot SNR}$, and θ is taken as the FWHM synthesized beam size. As mentioned above, the core component is likely blended with the inner jet emission. In reality, the peak emission of the core can be affected by intrinsic changes of the emission distribution (for example, the ejection of a new jet component or a passing shock). This might affect the consistency between the true intensity distribution and a fitted Gaussian component. All these factors may introduce an additional uncertainty of the reference point. We have fitted the core with several different models: a single elliptical Gaussian; one circular Gaussian components (the core) plus a point source (to represent the residual emission from the unresolved inner jet); two circular Gaussian components; two point sources. The discrepancy between the fitted core positions is used to estimate the systematic error of the reference point $\sigma_{R,sys}$, which is added into the total error budget.

The positional error of jet component can be expressed as $\sigma_R = \sqrt{\sigma_{R,sta}^2 + \sigma_{R,sys}^2}$. We found that the positional error of the brighter component J1 is dominated by the systematic error, while the statistical error contributes a significant fraction to the positional error of J2.

The positional errors of J1 and J2 along different components are presented below.

The new calculations infer a proper motion of J2 with a higher significance of 3σ , 0.019 ± 0.006 mas/yr. The proper motion J1 is slightly changed to -0.006 ± 0.004 mas/yr, due to the nature of the jittering jet head at jet-ISM interface.

The details of the error analysis are presented in Appendix C. The derived parameters are updated in the main text.

(3) The authors do not propagate measurement errors adequately when reporting other derived quantities. Some of their measurements have relatively large uncertainties (e.g., proper motion), and so their conclusions may change if considering the errors. It may be very difficult to quantify all the errors, but they need to be considered seriously when drawing conclusions.

The proper motions have been re-calculated based on the updated error analysis. The new parameters are presented in Table A6.

The error propagation is applied to the derived physical parameter. In particular, the brightness temperature is estimated as:

$$\sigma_{T_b}^2 = \left(\frac{\partial T_b}{\partial S}\right)^2 + \left(\frac{\partial T_b}{\partial D_{maj}}\right)^2 + \left(\frac{\partial T_b}{\partial D_{min}}\right)^2$$

where, T_b is the brightness temperature, S is the core flux density, D_{maj} and D_{min} are the major and minor axis of the core, respectively. The errors of T_b have been updated in column 9 of Table A3 in the manuscript.

(4) The bulk Lorentz factor is reported to be >3.3 , only an upper limit. While this is consistent with previous estimations (e.g., Zhang et al. 2017, An & Romani 2018), it alone does not rule out a possibility of a powerful jet (as opposed to the claimed 'less

We argue for the relatively less powerful jet based on the values of the Lorentz factor, proper motion and jet luminosity being lower compared to those in typical lower redshift blazars (e.g. Lister et al., 2016, AJ, 152, 12) in 8th paragraph.

(5) For inferring the jet power and ISM density (Appendix H), the authors assume that the spectral index is 0 between 1.4 GHz and 15 GHz for J1 and/or J2. Although previous measurements of the radio spectrum seem to support this albeit a spectral break at ~10 GHz, the measurements are for C+J1+J2 (or C as it dominates) as compared to J1+J2. Is it reasonable to assume the same spectrum for J1+J2 when the variability of C+J1+J2 is different from that of J1 (and/or J2) in Table 3 and Figure A1? This may be a concern when extrapolating the 15-GHz flux density to 1.4-GHz. In particular, the authors mention a propagating shock. The propagating shock may change the radio spectral shape in time (e.g., Valtaoja et al. 1992), hence assuming a radio spectral index of 0 in 2017-2018 based on earlier data needs a justification. In this regard, (near) contemporaneous multi-frequency observations may be helpful.

We agree with the referee that simultaneous multi-frequency measurements are crucial for accurate estimates of the spectral index. In addition to the spectral properties expected based on the turnover argument, this issue is now addressed in Appendix I on the basis of simultaneous multi-frequency measurements reported in Romani, 2006 (ref. 15)

(6) The empirical relation (between L_{jet} and $P_{1.4 \text{ GHz}}$) they used in Appendix H may have a relatively large uncertainty (a few orders of magnitude scatter in the data used for deriving the relation; Birzan et al. 2008); this (and other) uncertainty needs to be considered when deriving the jet power and claiming that it is low.

New estimates are made using the empirical relations presented in Foschini (2014) in Appendix I.

(7) The reported number $n \sim 65 \text{ cm}^{-3}$ (in the text and Appendix H) for the ISM density is too precise compared to the arguments given in Appendix H (see also comments 3-5). Shouldn't it be a limit as is for the density contrast $\rho_{\text{ISM}}/\rho_{\text{jet}} > 11$ because beta h is only a limit?

Agreed and corrected accordingly in Appendix I. A new number $n_e \geq 1.3 \text{ cm}^{-3}$ is calculated.

(8) It is said "In the highest-resolution images, a sharp (>90 degree) jet (bending is seen..." in the text. As the jet bending is seen in a previous study (of the same source (e.g., Zhang et al. 2017), the previous work needs to be (recognized).

The reference (Zhang et al. 2017) is added in 5th paragraph.

(9) It is said "a prominent X-ray emission (with a strong contribution from the disk).." in the text. It is believed that X-rays in these high-z blazars are (produced mainly by the synchrotron-self-Compton process and that disks of (supermassive blackholes mostly radiate in the optical band. Hence, the (statement needs to be clarified.

Agreed. The words '...with a strong contribution from the disk' are removed.

(10) Contours in Figure 1-e and Figure 2 are slightly different although it (appears that they are generated from the same observational data.

Figure 2 has been reproduced to use the same resolution with Figure 1e.

(11) In Appendix H, it appears that the authors consider J_1+J_2 . This should be mentioned explicitly. Assuming J_1+J_2 , I do not reproduce $S_{15\text{ GHz}}=21\text{ mJy}$ from the table. In 2017-2018, the average flux of J_1 and J_2 is $\sim 20.4\text{ mJy}$. The difference is only tiny, but I wonder if there is something else I am missing (here).

Corrected to $S_{15\text{ GHz, core}} = 90.94 \pm 5.18\text{ mJy}$ as applicable to the empirical relations requiring the core flux density in the calculation of the jet power.

(12) In Appendix G above equation (8) it is said "Employing equation (7) which reduces to...". Equation (7) does not seem to reduce to (8), so this may be a mistake.

Indeed, a mistake. Corrected.

Reviewer #3 (Remarks to the Author):

The essay deals with the changes in the jet structure from high-resolutions VLBI radio observations of the jetted AGN J0906+6930 ($z=5.47$). Authors find changes in the jet direction and interpret them as the interaction with a dense interstellar medium, suggesting a young age for this AGN. This topic is important and could be of interest for a larger public, as it addresses the presence of supermassive black holes (SMBH) in a relatively young universe. Since there is no time enough to form a SMBH via a sequence of mergers of smaller black holes, the detection of SMBH in the early universe sets the question of how they are formed. The essay could be published, but after some points that must be addressed by the authors. Specific comments follows:

- Authors indicated J0906+6930 as the most distant known blazar, having $z=5.47$. However, there is a recent detection by Saxena et al. (2018, MNRAS, 480, 2733) of a jetted AGN at $z=5.72$. The radio spectral index suggests it could be a radio galaxy, thus with the jet viewing angle greater than $\sim 10^\circ$. Therefore, the authors statements are formally correct. However, I would say a few words about this other jetted AGN.

Related sentences are added on the last sentence in the 7th paragraph of the main text.

- page 1, row 19: Authors wrote: "As jetted quasars seem to harbour heavier black holes than non jetted ones, high-redshift blazars are probes...". The line of reasoning is inverted. High-z quasars are useful probes for the early universe because they are the brightest sources, and, therefore, it is easier to find them. But, the extreme brightness is due to: (1) the presence of a relativistic jet viewed by a small angle, which means that the electromagnetic emission is Doppler boosted (with a typical Doppler factor of 10, the luminosity of the jet is boosted to 10000 times its isotropic value); (2) as the jet power scales with the mass of the central black hole (P proportional to $M^{1.4}$, cf Heinz & Sunyaev 2003), a larger mass implies a larger observed jet power.

The logic of the corresponding statement corrected in 2nd paragraph.

- page 1, row 28-29: the radio loudness is a meaningless parameter. No need to cite it.

The radio loudness is removed in 3rd paragraph.

- Section F: Doppler boosting parameter of the jet. Authors estimated the Doppler factor δ by using the brightness temperature as measured from the VLBI core size. The apparent jet speed β_{app} is measured from VLBI observations. From δ and β_{app} , it is then possible to calculate the bulk Lorentz factor Γ and the viewing angle θ . As the authors reported also the OVRO 15 GHz radio light curve, I suggest to calculate δ also from radio variability. An easier way is to refer to Liodakis et al. (2018, ApJ, 866, 137), who already did it and reported $\delta=8.91$. Together with $\beta_{app}=2.2c$ measured by the authors, it results $\Gamma=4.8$ and $\theta=3^\circ$. Please note that Liodakis calculated the equipartition brightness temperature is a way different from usual. So, when comparing your results with him, please take into account this difference. Please note that also the main text -- not only Section F -- must be changed accordingly.

Following the reviewer #3's very constructive suggestion, we added the calculation of variability Doppler factor in Appendix F. The monitoring data at 15 GHz from the OVRO 40-m telescope are used for calculating the variability brightness temperature ($T_{b,var}$). Liodakis et al.(2018) have decomposed the light curves into a series of flares characterized by an exponential rise and decay over and above a stochastic background. As a simplification owing to the light curve characterized by a single slowly flaring event, we modeled it with a Gaussian function

$$S(t) = A \exp\left(-\frac{(t - t_p)^2}{2\tau^2}\right) + B$$

where $S(t)$ is the source flux density (Jy), t is the observation time (days), B is the constant noise background (Jy), t_p is the peak time (days), and τ is the characteristic flare rise timescale (days). We used a least squares fit based estimation of the Gaussian parameters. The below figure shows the OVRO light curve (blue circles) and the fitted curve (red). The fitting gives: $t_p = 2012$ May 20, $\tau = 847$ days, $A = 0.149$ Jy, $B = 0.061$ Jy. Using Eq. (3) in [Hovatta et al.(2009)] to calculate the variability brightness temperature

$$T_{b,var} = 1.548 \times 10^{-32} \frac{S_{max} d_L^2}{(1+z)\nu^2 \tau^2},$$

where ν is the observing frequency (GHz), z is the redshift, d_L is the luminosity distance, S_{max} is the peak flux density and τ is the flare rising timescale (days) and is $T_{b,var} = 5.7 \times 10^{12} K$. The variability Doppler factor is

$$\delta_{var} = \left(\frac{T_{b,var}}{T_{b,eq}}\right)^{1/3}$$

corresponding to $\delta = 4.8$ which is consistent but moderately lower than the value of 6.1 derived from VLBI model fitting in Appendix F.

- Section H: authors calculated the jet power by using a relationship by Birzan et al. (2008). These relationships links the extended radio emission measured in the MHz frequency range with the jet kinetic power. Therefore, the application to one component detected at 15 GHz is rather risky, as it is known that extrapolating from high to low frequency could give serious errors in the estimation of the extended emission. In addition, the jet power by Birzan refers only to the kinetic component,

and does not include the radiative part. The kinetic part calculated in this way is based on the cavities excavated by the jet activity during million of years and are not comparable with a year time scale as from the authors of the present essay. I suggest to adopt the relationships by Foschini (2014, Int. J. Mod. Phys. Conf. Series, 28, 1460188), which links the radiative and kinetic power to the 15 GHz core luminosity. You should apply them to the core flux, but you can also try to apply them to the component J1. Obviously, you have to address the proper caveats. A quick calculation by using the core emission, resulted in a $\log P_{\text{rad}}=45.4$ [erg/s] and $\log P_{\text{kin}}=46.1$ [erg/s], greater than the estimates from the Birzan relationship, as expected.

We use the relations presented in Foschini (2014) for the revised calculations in Appendix I.

- the last (but not least) point refers to the estimation of the mass of the central black hole and the accretion rate. Authors refer to previous works, which gave a mass of the order of $10^9 M_{\text{Sun}}$ and accretion luminosity 0.4 times the Eddington limit. However, all the previous estimates adopted relationships made for non-jetted AGN. The presence of the Doppler boosting alters significantly the continuum emission, which implies an overestimation of the mass. Also the fit of the accretion disk suffers the same problem. As the optical spectra all show a strong Ly emission line, I suggest to use it together the relationships by Pian et al. (2005, MNRAS, 361, 919), which take into account the Doppler boosting. Indeed, a quick calculation of the $L(1350\text{\AA})$ by using the Pian's relationships results in a value of 7.4×10^{43} erg/s to be compared with the value of 5×10^{47} erg/s given by Romani (2004). The radius of the broad-line region (BLR) from Bentz et al. (2013), results to be 18.6 light days, which implies a luminosity of the disk of 3×10^{43} erg/s. I've not found a measurement of the FWHM of Ly, but if we assume the value of 5000 km/s from other lines, the mass of the central black hole results to be $7 \times 10^7 M_{\text{Sun}}$. Therefore, I suggest to recalculate the mass and accretion rate of the central black hole by using the Ly line parameters and Pian's relationships. Then, revise the discussion in the text according to the new values. A smaller mass of the central black hole would be more suitable for a young AGN, and also more consistent with a high-frequency peaker (HFP). Compare also with Berton et al. (2016, A&A, 591, A98).

The black hole mass is now revised. The discussion in text has been modified accordingly to reflect this with the addition of a new Appendix H.

Reviewers' comments:

Reviewer #1 (Remarks to the Author):

The authors have carefully considered the comments of the various referees and have made amendments that materially improve the paper. I would say that this work certainly deserves to be published somewhere and the revision work that the authors have done will therefore be useful. I am, however, not confident that this is the right venue. In the end re-reading the paper it seemed that the 'discovery of polarization' is not in and of itself surprising enough to merit a Nature communications publication. A detailed measurement of the polarization properties and exploitation of these to constrain the source physics would be very valuable. But that is for the future.

It seems that the various arguments about the jet dynamics are the real heart of the paper. I actually found these more interesting, although I did not agree with all conclusions. However these were too weakly justified at this point to make this paper very influential. Nor is it clear whether the authors' conclusions teach us much about high redshift jets, as a class. In sum, I think this paper is evolutionary rather than revolutionary. If the editors and other referees endorse publication, fine, but -- after this second reading -- I would advocate submission to a different Journal.

Reviewer #2 (Remarks to the Author):

The revised draft and the authors' response address the issues raised by the referees well. The revision is written clearly, and significant findings of this work, measurements of the radio polarization and the proper motion, are justified well.

However, there seem to be a few things to be further clarified (see below). So I recommend that this article is accepted for publication after clarifying the issues.

(1) The estimation of the BH mass is now lowered to $\sim 10^7 M_{\text{sun}}$ from $\sim 10^9 M_{\text{sun}}$. Implications of this on the SMBH growth in the early Universe need to be discussed.

(2) It is said that in J0906+6930 a nascent jet is embedded in a "dense medium". However, the density of the ambient medium is estimated to be >1.3 , which doesn't seem to be dense. This should be clarified.

(3) In appendix C and the author's response (page 4), it appears that the authors took a quadratic "mean" of the statistical and the fitting error. It is not clear why "mean" not "sum"?

(4) In the author's response (page 5), the error formula for T_b is strange; shouldn't it be

$$\sigma^2_{T_b} = (dT_b/dS)^2 dS^2 + (dT_b/dD_{maj})^2 dD_{maj}^2 + (dT_b/dD_{min})^2 dD_{min}^2 ?$$

If so, I am afraid that the error reported in the article is wrong.

(5) For the jet power estimation based on the radio luminosity, they used an empirical formula. In general, the data used for deriving the empirical formula (e.g., those in Foschini 2014) have large scatter (an order of magnitude) which is not explained with the formula. So simply accounting for the parameter errors in the empirical formula may be an underestimation of the uncertainty of the jet power. Although it is hard to quantify the scatter and doing so may not change the conclusion of the paper, the scatter should be discussed qualitatively at least.

(6) The authors say "A prominent X-ray emission indicates a high accretion rate, characteristic of a luminous optically-thick geometrically-thin disk". It is still not clear how X-rays are related to accretion or a disk in these SMBHs (not stellar-mass BHs). So this should be explained. If the authors would like to argue for a strong disk (or accretion) in the source, it may be better to say strong disk emission in the optical band seen in high- z blazars; see Romani (2016) for J0906+6930, and An & Romani (2018) for four high- z blazars.

Reviewer #3 (Remarks to the Author):

The revised manuscript is fine for me. I warrant the publication.

Authors' reply on the second round of reviewers' comments on
Evolving parsec-scale radio structure in the most distant blazar known
by T. An et al.

We thank the reviewers for the second round of comments. We have taken them into account, and hope that it improves the manuscript. Changes made in the main text are shown in blue fonts. Our responses to the second round of comments (in straight fonts) are shown below in italics.

Reviewer #1 (Remarks to the Author):

The authors have carefully considered the comments of the various referees and have made amendments that materially improve the paper. I would say that this work certainly deserves to be published somewhere and the revision work that the authors have done will therefore be useful. I am, however, not confident that this is the right venue. In the end re-reading the paper it seemed that the 'discovery of polarization' is not in and of itself surprising enough to merit a Nature communications publication. A detailed measurement of the polarization properties and exploitation of these to constrain the source physics would be very valuable. But that is for the future.

It seems that the various arguments about the jet dynamics are the real heart of the paper. I actually found these more interesting, although I did not agree with all conclusions. However these were too weakly justified at this point to make this paper very influential. Nor is it clear whether the authors' conclusions teach us much about high redshift jets, as a class. In sum, I think this paper is evolutionary rather than revolutionary. If the editors and other referees endorse publication, fine, but -- after this second reading -- I would advocate submission to a different Journal.

We absolutely do not claim that the presented results are revolutionary. They certainly might be treated as evolutionary. However, our intention is to present the novel results (the first of the kind on some accounts) and provide a plausible scenario to interpret the observations. These are:

1. The detection of polarization in a high- z quasar ($z > 4.5$) is rare and the associated calibration is challenging. As explained in the manuscript, (i) the project focused originally on the total intensity imaging with a rather short total on-source time, and (ii) the source flux density was in a declining state. Yet, not only the original goal of intensity imaging was achieved, but the compact polarised emission was detected and localised too. .

2. Of particular note here is that the observed frequency of 15 GHz corresponds to ~ 100 GHz in the quasar's rest frame, thus allowing for a probe well inside the optically thin jet base. As the detection of polarisation is in the jet at an appreciable distance from the core region, it is indicative of regular magnetic field. Detection of polarised emission at such the high rest frame frequency is rare if not unique so far.

3. Determination of high- z jet proper motion is also challenging due to the cosmological time dilation effect, requiring long time-span VLBI monitoring and a suitable target with compact core-jet structure. 0906+6930 is one of few satisfying the requirements, and it is at the highest redshift. The present work suggests a moderate relativistic jet in this source.

A detailed measurement of the polarisation distribution in this jet region can be attempted with follow up observations when the source becomes brighter. This will help in further

probing the proposed scenario. Our result will serve as a starting point for future detailed investigation.

Reviewer #2 (Remarks to the Author):

(1) The estimation of the BH mass is now lowered to $\sim 10^7 M_{\text{sun}}$ from $\sim 10^9 M_{\text{sun}}$. Implications of this on the SMBH growth in the early Universe need to be discussed.

The revised mass of the SMBH is consistent with the expectation from typical high-frequency peaked quasars. A detailed comment on the SMBH growth in the early Universe requires an understanding of the accretion process and fuelling mechanisms, as well as an AGN feedback through radiation and jet interaction with the large scale environment in addition to the effects of galaxy mergers. These can be better addressed with a larger sample of high redshift sources and multi-wavelength observations, amended by simulations. All these prospective steps are beyond the scope of the present work. We have included a short sentence on the implications and some context in the main paper 7th paragraph and in section H.

(2) It is said that in J0906+6930 a nascent jet is embedded in a "dense medium". However, the density of the ambient medium is estimated to be >1.3 , which doesn't seem to be dense. This should be clarified.

The dense medium refers to the density contrast of >9 between the surrounding interstellar medium (ISM) and the jet, as obtained just outside the jet head. The number density is now revised to be $> 26.6 \text{ cm}^{-3}$ after properly accounting for the area in contact at the jet head - ISM interface. The discussion in text is slightly modified in Appendix Section I to clarify the context and reflect these changes.

(3) In appendix C and the author's response (page 4), it appears that the authors took a quadratic "mean" of the statistical and the fitting error. It is not clear why "mean" not "sum"?

We are using the standard error propagation method to estimate the total error. The formula/method adopted is from Chapter 3 of "Propagation of Errors" (by Mike Peralta, pressed by CreateSpace Independent Publishing Platform).

(4) In the author's response (page 5), the error formula for T_b is strange; shouldn't it be $\sigma_{\langle T_b \rangle} = \left(\frac{dT_b}{dS}\right)^2 dS^2 + \left(\frac{dT_b}{dD_{\text{maj}}}\right)^2 dD_{\text{maj}}^2 + \left(\frac{dT_b}{dD_{\text{min}}}\right)^2 dD_{\text{min}}^2$? If so, I am afraid that the error reported in the article is wrong.

The mentioned formula in our response had a typographical error. We apologise for this. We have used the above mentioned correct error propagation relation in the main text to estimate the errors which are quoted in text.

(5) For the jet power estimation based on the radio luminosity, they used an empirical formula. In general, the data used for deriving the empirical formula (e.g., those in Foschini 2014) have large scatter (an order of magnitude) which is not explained with the formula. So simply accounting for the parameter errors in the empirical formula may be an underestimation of the uncertainty of the jet power. Although it is hard to quantify the scatter and doing so may not change the conclusion of the paper, the scatter should be discussed qualitatively at least.

This is now discussed in section I. A new sentence is added: "It must be noted that the empirical relations have a large scatter mainly due to the diversity in the sources constituting the inference, ranging from radio-loud AGN to X-ray binaries, and having to properly account for spectral state transitions, variability and systematics from comparison of data from different databases⁵². The uncertainty on the jet power may then be underestimated."

(6) The authors say "A prominent X-ray emission indicates a high accretion rate, characteristic of a luminous optically-thick geometrically-thin disk". It is still not clear how X-rays are related to accretion or a disk in these SMBHs (not stellar-mass BHs). So this should be explained. If the authors would like to argue for a strong disk (or accretion) in the source, it may be better to say strong disk emission in the optical band seen in high-z blazars; see Romani (2016) for J0906+6930, and An & Romani (2018) for four high-z blazars.

A corresponding modification is now made and a sentence is added in the second last paragraph.: A prominent disk emission is inferred from the modelling of the spectral energy distribution of this and three other high-z blazars²⁷.

Once again, we thank the referees for the constructive comments and suggested improvements of the manuscript.

REVIEWERS' COMMENTS:

Reviewer #2 (Remarks to the Author):

I find that the revision resolved and clarified some issues. The authors seem to properly interpret the proper motion and polarization measurements for this highest- z blazar as due to nascent jets interacting with dense ambient medium, and present their results and arguments clearly.

These are certainly worth a publication. However, unknown uncertainties in the measurements, the previous models used for interpolating their results, and the inferred physical properties weaken the arguments of the authors. Furthermore, there are possibilities of alternative scenarios, and therefore I am not convinced that the results significantly influence my thoughts on SMBHs and their evolution in the early Universe. The results help to strengthen previous arguments on the subject and so I think that the paper fits into another journal better.

REVIEWER COMMENTS:

Reviewer #2 (Remarks to the Author):

I find that the revision resolved and clarified some issues. The authors seem to properly interpret the proper motion and polarization measurements for this highest- z blazar as due to nascent jets interacting with dense ambient medium, and present their results and arguments clearly.

These are certainly worth a publication. However, unknown uncertainties in the measurements, the previous models used for interpolating their results, and the inferred physical properties weaken the arguments of the authors. Furthermore, there are possibilities of alternative scenarios, and therefore I am not convinced that the results significantly influence my thoughts on SMBHs and their evolution in the early Universe. The results help to strengthen previous arguments on the subject and so I think that the paper fits into another journal better.

Thanks for the referee's comments. The implications of AGN jet activity on SMBH growth in the early Universe, and additional alternative scenarios enabling the jet structure are referred to in the 9th paragraph in section 'Results and Discussion' and discussed in Methods.